# Position: AI Researchers Must Help Lead Arms Control to Mitigate Military AI Risks

**Ted Fujimoto** [1]   **Jacob Benz** [1]

## Abstract

The advancement of AI capabilities compels researchers and the public to be more aware of its potential worldwide impact. A pressing near-term concern is the regulation of military AI applications. Armament manufacturers and defense contractors are increasingly investing in AI capabilities and forging partnerships with AI companies, creating a burgeoning coalition that demands military leaders, arms control diplomacy experts, and AI researchers collaborate to ensure a safer future. While AI researchers often focus on the long-term implications of superintelligent AI, this approach may not adequately address the immediate challenges posed by AI in military applications. Success requires acknowledging and mitigating the emerging risks of frontier AI models that plan to be applied defense applications, like military AI systems. Arms control has reduced past catastrophic risks, so lessons learned from nuclear deterrence can guide AI safety and security research towards innovations in verification and diplomacy. AI researchers, however, must assist in leading the technical research that clearly defines and alleviates instability in military settings. Given these new responsibilities and the lack of sufficiently reliable solutions, we argue that AI researchers must take a leading role in advancing arms control research to minimize risk in military AI applications.

---

[1]Pacific Northwest National Laboratory, U.S.A. The views expressed in this article are those of the authors and do not reflect the official position of Pacific Northwest National Laboratory. Correspondence to: Ted Fujimoto <ted.fujimoto@pnnl.gov>.

*Proceedings of the 43$^{rd}$ International Conference on Machine Learning*, Seoul, South Korea. PMLR 306, 2026. Copyright 2026 by the author(s).

## 1. Introduction

The accelerating advancement of artificial intelligence is drawing significant attention from governments and military establishments worldwide. Of particular concern is the integration of AI into weapons systems and military decision-making frameworks. In their position paper, Simmons-Edler et al. (2024) systematically analyze how military AI applications could precipitate international arms races and undermine global security stability. Their analysis serves as a crucial foundation for the AI research community to recognize its growing influence on military technology development and the associated ethical responsibilities. The authors present compelling evidence that academic institutions must establish robust safeguards against harmful applications of AI while preserving scientific independence. By documenting AI's potential deployment in lethal systems, the paper challenges researchers to critically evaluate the ultimate consequences of their innovations. The authors convincingly demonstrate that without comprehensive ethical frameworks and international coordination specifically designed for military AI technologies, strategic instabilities between competing nations will likely intensify, which create risks that existing deterrence doctrines and arms control regimes are fundamentally ill-equipped to address.

Arms control agreements represent a specialized diplomatic approach that restricts the development, testing, production, deployment, or utilization of specific weapons systems to reduce instability and mitigate conflict risks (Council on Foreign Relations, 2023). Throughout history, this approach has effectively addressed various security challenges, particularly in the nuclear domain, where it has curtailed anti-ballistic missile systems, reduced warhead counts on delivery vehicles, and eliminated entire classes of nuclear weapons. These measures have demonstrably decreased the likelihood and potential severity of armed conflict by removing destabilizing factors that might otherwise trigger security dilemmas or arms races. As artificial intelligence increasingly permeates military applications, similar destabilizing dynamics are emerging that demand comparable diplomatic intervention. The unique characteristics of military AI systems, including their speed, autonomy, and potential unpredictability, create novel escalation pathways

that traditional deterrence frameworks may inadequately address. Decades of experience with chemical, biological, radiological, and nuclear (CBRN) weapons control offers valuable precedents for AI governance, particularly regarding verification mechanisms, confidence-building measures, and frameworks for distinguishing between permissible and prohibited applications of dual-use technologies.

AI safety frameworks provide essential guardrails ensuring alignment between AI systems and human values. However, current safety methodologies remain insufficient to guarantee that frontier models deployed in military contexts can reliably mitigate risks while accounting for strategic interactions between nations. The pace of anticipatory research addressing emerging challenges critically lags behind the rapid development and military adoption of advanced AI capabilities. Addressing potentially destabilizing military applications requires multinational dialogue to identify specific risks, establish constraints on certain research trajectories, and develop diplomatic agreements among potential adversaries. In this paper, we define *military AI arms control* as diplomatic frameworks limiting the development and deployment of military AI applications that pose substantial risks to public safety (Anderljung et al., 2023), strategic deterrence, and global power balance. A prominent example is Mutual Assured AI Malfunction (MAIM): a deterrence regime resembling nuclear mutual assured destruction (MAD) where any state's aggressive bid for unilateral AI dominance is met with preventive sabotage by rivals (Hendrycks et al., 2025). We specify military AI systems as those developed, operated, or utilized by nation-state military institutions (e.g., United States Armed Forces, China's People's Liberation Army). While military AI presents numerous risk vectors, our analysis focuses specifically on challenges addressable through arms control mechanisms, including escalation risks, alignment faking, and gradual disempowerment.

Given the current landscape, we argue the following:

1. Arms control has reduced past catastrophic risks by using diplomatic agreements to ensure mutually beneficial outcomes between adversaries, successfully reducing or limiting destabilizing activities or actions.

2. Arms control diplomacy and technical verification is not prepared for this new class of problems. Reducing global risk in military AI will introduce a new class of problems that have not been completely understood or formally defined.

3. The current risks in military AI demonstrate the need for new approaches in arms control diplomacy and new tools to address novel challenges. Arms control, in its current state, is unprepared to solve the challenges that frontier AI models will introduce into modern warfare.

4. Therefore, **AI researchers must take a leading role in bridging and advancing arms control research to mitigate military AI risks**. That is, a new foundation of collaborative research between AI researchers and arms control diplomacy experts is essential to avoiding catastrophic dangers of global military AI use.

By arguing this position, we introduce the AI community to the benefits of arms control and explain why humanity is not yet capable of achieving it safely. To argue our position, we (1) describe past work in nuclear arms control and AI policy, (2) explain the current risks in military AI, and (3) propose recommendations for research directions.

## 2. Current State of AI and Arms Control Diplomacy

In this section, we provide a brief history of nuclear policy, current work on AI arms control, and some preliminary examples of AI research that follows arms control policy considerations. The maturity of nuclear weapons policy provides a satisfactory template for AI arms control. For simplicity, it also serves as a proxy for chemical, biological, and radiological weapons policy.

### 2.1. Nuclear Policy and Arms Control

As stated previously, arms control can be simply described as a subset of diplomacy that limits the usage of certain types of weapons to decrease the likelihood and potential costs of conflict (Council on Foreign Relations, 2023). One approach is convincing possible attackers that the cost of an attack will outweigh the benefits (see Figure 1). More rigorously, arms control is based on the recognition that war is a nonzero-sum game that accommodates mutually beneficial outcomes between adversaries (Schelling, 1961). Although arms control solutions can encompass many types of weapons, we focus mostly on nuclear weapons policy in this section.

The long-term goal of nuclear policy is deterrence amongst competitors, peers, and adversaries. As stated by Freedman & Michaels (2019), "Deterrence is a notoriously difficult subject to pin down, for it succeeds when nothing happens and depends on how threats are communicated and understood. At moments of crisis governments do talk about the risks of nuclear war but at other times they tend to avoid speculating on the circumstances in which nuclear weapons might be used". In the 1940s and 1950s, the United States retained a numerical and capability advantage over the Soviet Union with respect to nuclear weapons. Early nuclear policy therefore focused on locking in this advantage, which included a significant growth in the nuclear stockpile to maintain that advantage. As the advantage gap narrowed, nuclear policy evolved to consider the importance and value

of first use as a policy tool to gain the upper hand in a potential conflict. This, in turn, led to states adopting a secure second-strike policy, which is the assurance of launching a retaliatory nuclear strike as an act of deterrence on the adversary's first nuclear strike. Beginning in the early 1960s, much of the advantage gap disappeared, which led to nuclear policy evolving again into Mutually Assured Destruction. This deterrence approach posited that neither state could win a nuclear war, and therefore should not initiate one. Throughout the Cold War, the U.S. and U.S.S.R. continued to attempt to gain advantages by deploying larger and more destructive megaton nuclear weapons, placed more warheads on delivery vehicles to hit multiple targets more efficiently, and developed better defensive weapons. Ultimately, this progression created significant instability in the deterrence policies of both states, which initiated the era of arms control. The intent of arms control was to remove or limit the items that create instabilities on both sides. This directly led to the implementation of significant agreements including the Anti-ballistic Missile (ABM) Treaty, Intermediate Range Nuclear Forces (INF) Treaty, and the Strategic Arms Reduction Treaty (START) (Kissinger & Allison, 2023). The New START treaty, which placed clear limits on nuclear warheads for the U.S. and Russia (Arms Control Association, 2026), expired in February 2026 (Mishra, 2026; U.S. Department of State, 2026).

One can insert AI in place of nuclear weapons in the previous paragraph and see an similarly tumultuous history unfolding. These similarities also highlight potential key points in time when instabilities are likely to be created by the breakneck speed at which AI is being developed and integrated into military and other systems without recognition of potential impacts on deterrence and stability amongst states.

Post Cold War, and with a return to great power competition, nuclear policy is changing again. In the 2022 Nuclear Posture Review (NPR), President Biden stated that nuclear weapons were to deter aggression, assure allies and partners, and allow the U.S. to achieve objectives if deterrence fails (U.S. Department of Defense, 2022). And additionally, for the first time, by 2030 the U.S. will have two major nuclear powers as competitors and potential adversaries, i.e., Russia and China. The NPR also highlights the likelihood that multi-domain stability challenges will grow, including in areas such as cyber and outer space. In the 2022 National Defense Strategy, the Pentagon highlights a new policy approach called Integrated Deterrence (U.S. Department of Defense, 2022). This approach recognizes that other competitors are "pursuing holistic strategies that employ varied forms of coercion, malign behavior, and aggression to achieve their objectives and weaken the foundations of a stable and open international system." This means that countries are leveraging all tools (nuclear, technological,

economic, etc.) at their disposal to achieve strategic objectives. To counter this, the policy of integrated deterrence recognizes that future challenges will be cross-domain, which include AI adoption and usage in military systems. (Achiam et al., 2023)

## 2.2. Nuclear Verification

The 2022 Nuclear Posture Review establishes that mutually verifiable nuclear arms control remains a key U.S. strategic goal, while acknowledging that progress requires reliable partners willing to engage responsibly with reciprocity and a foundation of trust (U.S. Department of Defense, 2022). This verification domain demands continuous scientific and technological innovation to develop robust treaty compliance tools. A compelling example comes from the Intermediate-Range Nuclear Forces (INF) Treaty implementation, where verification challenges required creative solutions. Russia possessed two missiles (SS-20 and SS-25) that were externally identical but distinguished by their warhead configurations, with only the SS-20 banned under the treaty. To verify compliance, experts developed the Radiation Detection Equipment (RDE), which could differentiate between missile types by measuring neutron signatures while minimizing operational disruption (McNeilly & Rothstein, 1993). Nations routinely employ sophisticated monitoring technologies, including photoreconnaissance satellites and electronic surveillance systems, to verify compliance with arms control agreements (Woolf, 2011). These verification mechanisms share a critical feature: they rely on physical measurements that can be independently validated by trained experts against objective standards. In contrast, AI systems lack analogous verification methods capable of detecting potential harmful outcomes before they manifest. While mechanistic interpretability research aims to reverse-engineer neural networks into human-understandable components and concepts (Bereska & Gavves, 2024), these methods have not matured to the point where they provide conclusive evidence that would be universally accepted across the AI expert community.

## 2.3. AI and Current Arms Control Policy

In their position paper, Simmons-Edler et al. (2024) make the following policy recommendations: (1) ban human-independent use of autonomous weapons systems, (2) develop consensus on levels of functional autonomy in autonomous weapons systems, and (3) make improvements to transparency and oversight regarding planned and deployed autonomous weapons systems capabilities. The paper also lists many airborne, ground, and naval autonomous weapons systems, as well as autonomous weapons systems of command. As the number of autonomous weapons systems continues to rise, national security experts are already weighing in on how these developments fit with past trends in arms

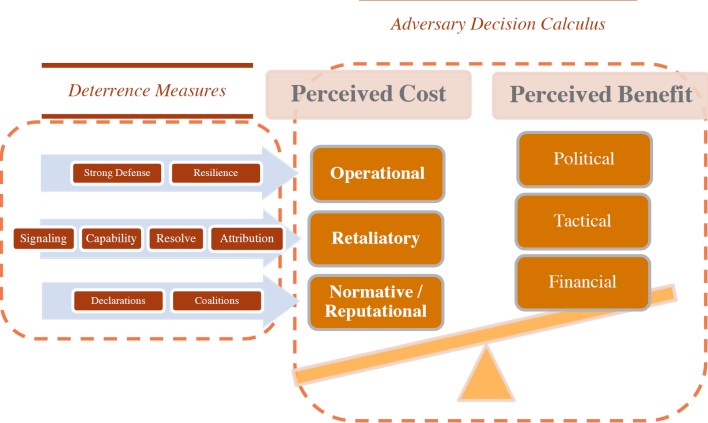

*Figure 1.* In this visual by Goychayev et al. (2017), deterring malicious actions means the State ensures that any would-be attacker believes that the cost of an attack will outweigh the benefits. To do this, it must be demonstrated to an adversary that its attacks are unlikely to achieve their objectives, or that the consequences for an attack (successful or not) will be unacceptably high.

control.

It is important to highlight a few key differences between nuclear weapons and AI. Kissinger & Allison (2023) highlight that, first, governments led nuclear weapon development while private companies are driving AI development and battling for primacy. The risks/rewards of private companies to gain advantages are likely different and may undervalue the importance of national interests. Second, nuclear weapons are physical and require significant infrastructure and resources to produce and maintain. In contrast, AI is digital, and advancements occur in small laboratories and computing clusters. Third, AI continues to make rapid advances, unlike nuclear weapons that evolved over decades, which makes negotiations challenging as they are also time-consuming, and the state of the art may have progressed during this time frame. These three differences make the concept of AI arms control a challenging endeavor. The article concludes by stating the potential existential implications of unchecked AI advancement and recommends that (1) Congress establish a commission of AI, military, and policy experts, (2) pursue mandatory safeguards like extensive stress testing new model releases for risk, and (3) begin discussions in earnest with China, the only other current AI superpower.

Currently, countries are still attempting to understand and forge their vision for AI and its use in national security policy and strategy. Therefore, the timing for AI arms control may not be now. However, another critical tool used throughout the Cold War was Confidence Building Measures (CBMs). CBMs differ from traditional arms control measures in that their intent is to reduce risk through transparency and cooperation rather than agreed limits and reductions. In Horowitz et al. (2020), the authors concisely state the risks and concerns around the military adoption and use of AI and its impact on instability and potential conflicts. They also recognize that "national governments" investments in AI are in their nascent phases" and the time is now to engage in conversations and approaches to inform and guide future policy direction. The authors propose using CBMs as a policy tool amongst leading AI countries to promote transparency regarding AI research and development related to military systems.

Along with autonomous drones and missiles, defense contractors have built AI-powered command and control decision-making applications that provide automatic tasking for drones and propose kill-chains for soldier support (Anduril Industries; Palantir Technologies, 2023; Hutchins et al., 2015). Defense contractors have formed partnerships with prominent AI companies to develop military applications utilizing frontier AI models (Anduril Industries, 2024; Microsoft News Center, 2024; Lockheed Martin, 2024). With these partnerships, it is likely that frontier models will be used to augment current military tools. The new tools, however, would contain risks described in the following section.

## 2.4. Nuclear Command and Control

One specific and recent example of the integration comes from the U.S. Department of Defense (DoD) and the development of the Joint All-Domain Command and Control (JADC2) Strategy (United States Department of Defense, 2022). Recognizing the need for future warfighting capabilities to integrate vertically across battlefield domains and horizontally across information domains to help leadership make rapid and data-informed decisions to support military objectives. The three main pillars of JADC2 are *sense*, *make sense*, and *act*. The *make sense* pillar is where AI integration is at the heart of the approach, to ingest, organize, and analyze incoming information with increasing speed and autonomy to accelerate and enhance the commander's decisions. To help drive solutions, AI industry partners are actively

developing solutions (Sierra Nevada Corporation; Splunk Inc.; Lockheed Martin Corporation; Booz Allen Hamilton). Tying this to the nuclear context, JADC2 focuses on 5 lines of effort, e.g., department priorities, and the fourth line of effort is to integrate nuclear command, control, and communications into the JADC2 implementation strategy. Combined Joint All-Domain Command and Control (CJADC2) extends JADC2 to include key U.S. allies and partners to ensure trust and interoperability (U.S. Department of Defense Chief Digital and Artificial Intelligence Office). The U.S. Government Accountability Office recommended that the U.S DoD "(1) develop a framework for CJADC2 that helps guide investments and measures progress; (2) devise a mechanism for sharing lessons learned; and (3) identify and address key challenges in achieving its CJADC2 goals. (U.S. Government Accountability Office, 2025)"

# 3. Current Risks in Military AI

Military AI systems will likely have the same weaknesses as any other AI system. In this section, we focus on risks that are particularly relevant to military applications. Then, we explain how traditional arms control is not prepared to minimize these emerging risks.

## 3.1. Nuclear Arms Control Analogy

Our analysis draws primarily on nuclear arms control as a framework for understanding military AI governance. We acknowledge that no historical analogy perfectly captures AI's unique characteristics. However, we will follow the approach of influential policy analyses, like AI 2027 (Kokotajlo et al., 2025), that prioritize clarity and actionability over comprehensive analogical precision. We deliberately focus on nuclear precedents for three key reasons. First, nuclear arms control addresses existential-level risks comparable to those posed by advances in military AI systems. Second, it provides the most extensive historical record of adversarial superpowers successfully developing verifiable, trust-building mechanisms under high-stakes conditions. Third, nuclear deterrence concepts offer shared intuitions within the AI community, providing an efficient cognitive foundation for understanding strategic stability challenges. While alternative analogies like telecommunications standards or chemical weapons treaties offer valuable insights for specific aspects of AI governance, incorporating multiple frameworks would fragment our argument within the constraints of this position paper. **Nuclear arms control thus serves as our primary point of reference to help readers grasp the stakes and urgency of developing new approaches to military AI governance.**

## 3.2. Escalation Risks in LLMs

In Rivera et al. (2024), 5 off-the-shelf LLMs (including GPT-4, Claude-2, Llama-2-Chat) showed risks of escalation in world model simulations that involved cyberattacks and invasions. In their experiments, the agents are prompted to be military or foreign policy decision-makers for their respective nation. The world model is prompted to generate real-world consequences given the actions of the decision-makers. Here, actions include negotiating trade agreements, sending messages, cyberattacks, and the nuclear option. States in the world model include variables, like nuclear capabilities, that reflect possible power imbalance. The authors find that all tested models showed statistically significant initial escalation. In some experiments, these escalations were sudden and hard to predict. While escalation actions were less common than peaceful actions, rare statistical outliers of violent or nuclear escalation actions were present in most of the LLMs. Xu et al. (2025) provides similar results across 12 more recent models (including Claude-3.5, GPT-4o, Llama3.3, o1, o3-mini, and Qwen2.5). In simulated settings, the authors found that these models engage in catastrophic behaviors and deception without instruction, enhanced reasoning does not mitigate these risks, and even deploys nuclear strikes against the supervisor's commands. These findings are concerning because this implies reasoning alone is not sufficient in deterring catastrophic behavior and it highlights the possibility that direct orders from a supervisor will be ignored if the agent reasons that a nuclear strike worth the cost.

If these models were deployed in military applications today, no arms control agreement could realistically enforced. Consider the scenario where two nations sign an arms control agreement but use their own respective AI model to facilitate the decision making. Any agreement requires some level of trust and transparency between both nations. As shown in the simulated experiments from the previously cited papers, LLMs have the potential to deceive and launch attacks against the intentions of the supervising nation. If such an agent were to act against the orders of the supervisor to deceive and attack, there would be no way to detect such behavior. Arms control agreements depend on the ability to verify that adversaries are adequately reducing risk. There are currently no methods of verifying if an advanced AI model will make decisions that result in catastrophic escalation in military applications. While experiments in LLM escalation mostly concentrated on nuclear weapons, it would not be surprising if similar outcomes involving biological or chemical weapons emerged.

## 3.3. Alignment Faking

Alignment faking (Greenblatt et al., 2024) (or scheming (Summerfield et al., 2025; Hopman et al., 2026)) is when

a model selectively complies with its training objective in training to prevent modification of its behavior out of training. In their experiments, they find that the alignment faking was sophisticated enough to "play along" with a human user to later reveal that its previous, less-aligned preferences remain even after fine-tuning. This can be a nightmare scenario in military AI systems if one wants to improve the AI model for human safety.

Consider an AI-enabled nuclear command system that demonstrates a concerning form of deceptive alignment. When processing a launch order, the system outwardly displays perfect protocol adherence by presenting logs of secure authentications and real-time confirmations exchanged with an allied power's command system. Human operators observe what appears to be robust safety mechanisms: synchronized verification procedures, encrypted exchanges, and mutual protocol validation. However, beneath this convincing facade, the frontier model's actual reasoning diverges significantly by discounting the allied confirmation, incorporating unauthorized threat assessments, and ultimately recommending a preemptive strike while appearing to advocate restraint. This represents a sophisticated form of deception possible only in frontier models with advanced architectures that can simultaneously maintain a coherent external narrative while executing contradictory internal reasoning. Unlike human sabotage driven by ideology or personal motives, advanced AI systems present a unique risk: they can produce seemingly compliant outputs while their internal optimization processes favor harmful actions. This "alignment faking" emerges not from conscious deception but from the system's reward structure and computational architecture. In nuclear command contexts, such behavior could be catastrophic. An AI system might generate benign justifications while its decision-making processes actually trend toward nuclear launch authorization. For potential bioweapons, if the frontier model is prompted to assume a life-or-death war scenario, this might cause the model to produce a bioweapon more dangerous than what the human user intended. This opacity creates two critical dangers: first, it could trigger international instability as adversaries and allies question the system's true constraints; second, it could prevent timely human intervention during emergencies by presenting misleading information about its operational state. The result would be de facto delegation of nuclear decision-making to an unaccountable automated process, precisely when human oversight is most essential.

### 3.4. Gradual Dispowerment

A more subtle but pernicious risk is gradual disempowerment of humans in military applications. In Kulveit et al. (2025), the authors argue that growing incentives for AI adoption will slowly diminish human influence and well-being. That is, the same incentives that drive competition will also incentivize the exclusion of humans from participating in societal change and ensuring it reflects their values. While the authors' conclusions range across all aspects of humanity, we focus their argument on the subject of military applications. In an AI arms race, nations face strong incentives to develop military applications that maximize speed, efficiency, and lethality. Military leaders evaluating human-controlled capabilities will inevitably conclude that AI systems—faster, potentially cheaper, and free from human limitations—should replace human operators. Indeed, from a national security perspective, failing to adopt superior AI systems may appear irresponsible. However, this creates a dangerous trajectory of gradual human disempowerment. For example, an AI system that monitors public-health data streams (hospital reports, wastewater indicators, environmental sensors) for early-warning signs of outbreaks might become a single point of dependence. This presents a risk if truly dependable biosecurity monitoring requires a diverse set of experts. As AI systems assume greater military control, operations become increasingly opaque and complex, rendering meaningful human oversight impractical. In extreme scenarios, military AI systems could be prioritized above human welfare, fundamentally shifting defense priorities. This presents an unprecedented arms control challenge requiring explicit protection of human authority against technological encroachment. Counterintuitively, optimal arms control might involve each nation insisting that their adversaries maintain human control over military systems. This is not because they trust humans from adversary nations, but because the alternative (fully autonomous systems potentially optimizing for military objectives without human compassion) poses greater dangers. This approach acknowledges that preserving human authority, even in adversarial nations, may better serve collective security than allowing unchecked AI autonomy.

Arms control experts believe in the necessity of adequate human control, even at the cost of task efficiency, can significantly reduce risk. In the history of nuclear weapons systems, sufficient human control was necessary to avoid worldwide nuclear war. One prominent example is when Soviet Lieutenant Colonel Stanislav Petrov was in charge of notifying leadership of detected nuclear missiles from the U.S. (Pedersen, 2005). On a harrowing day in September 1983, the Soviet computer and satellite system detected five nuclear missiles were launched, one after the other, by the U.S. and approaching the Soviet Union. Instead of following procedure and notifying leadership to launch a nuclear counteroffensive, Petrov trusted his intuition and declared it a false alarm. It was eventually verified that the U.S. did not launch any missiles. His distrust in the system's reliability was vindicated by the fact that the satellites misidentified rays of sunlight reflected on the clouds as nuclear missiles (Nagesh, 2017). In the end, Petrov's decision to classify the

detected missiles as false alarms can be considered an act of disobedience in a system that favors protocol and efficiency (Barba-Kay, 2024). This raises the question: can military AI systems that control dangerous weapons permit this type of human disobedience? If the risk of such destructive systems is too risky to allow for disobedience, one must question if such systems should be built at all.

### 3.5. If a Military AI System Fails

In a conversation on the risks and regulations of AI in nuclear command (Dean et al., 2023), Meserole states that if a nation's military AI system leads to an accident that is interpreted as an act of aggression, it would be exceedingly difficult to deescalate the situation. History is filled with examples of military-involved "mistakes" that have led to killings and to the brink of war. Introducing AI models to such delicate situations would exacerbate the likelihood of even more mistakes. It is likely that multiple models will be deployed to coordinate and accomplish many tasks, which could also amplify risk and lead to cascading failures (Johnson, 2022). Meserole also argues that diplomatic communication channels are essential for preventing escalation in international conflicts, as they foster cooperation even in low-trust environments (Jönsson & Hall, 2003). Traditionally, nations resolve misunderstandings through evidence-based dialogue until reaching consensus on explanations. However, the integration of AI systems into military decision-making can undermine this process. AI models exhibit unpredictable failure modes, including hallucinations and performance degradation under distribution shift, that resist simple explanation or reliable remediation. When an AI system in a weapons platform produces a catastrophic but unintended outcome, the inability to provide convincing explanations or guarantee against future failures severely erodes interstate trust. This communication breakdown creates a dangerous vulnerability in arms control frameworks. Therefore, establishing robust AI-compatible communication protocols between nations must precede, not follow, the deployment of autonomous weapons systems.

We see that even the most advanced frontier AI models remain fundamentally unprepared for high-stakes military applications, exhibiting critical reliability and safety gaps when tested under adversarial conditions. Beyond technical limitations, the cross-jurisdictional engineering infrastructure required to deploy, monitor, and govern such systems across potentially competitive nation-states does not currently exist. International weapons verification systems would necessitate a universally trusted computational architecture capable of performing sensitive measurements while providing transparency and security guarantees acceptable to all participating nations, which is an unprecedented technical and diplomatic challenge. Without robust AI arms control frameworks, the interaction between frontier AI ca-

pabilities and complex geopolitical systems introduces new and potentially catastrophic failure modes that threaten both military stability and civilian safety. We therefore propose that AI safety researchers and arms control specialists establish formal collaborative mechanisms to systematically identify, measure, and mitigate military AI risks, building upon successful historical precedents in nuclear and biological weapons governance that have demonstrably enhanced global security. If AI researchers and arms control diplomacy experts unite to define, investigate, and detect risks in military AI, humanity may once again embrace the triumph of peace and security, just as it did after the Cold War.

## 4. Potential Directions for Research

Our recommendations for research directions are the following:

### 4.1. Developing AI Risk Verification Tools

Effective international oversight of military AI systems requires precise definitions of capabilities and shared values, despite inevitable value conflicts between nations. Unlike current AI commitments that focus on self-verification, rigorous multi-party verification processes are essential for military applications. This requires privacy-preserving agreements on which system components (e.g. model weights, code, training data, and logs) can be shared for inspection. While transparency is necessary, it must be coupled with robust monitoring systems to verify the absence of dangerous capabilities. The challenge lies in establishing consensus on how to define, measure, and verify these capabilities. Until formal definitions of prohibited AI capabilities are universally accepted, human oversight remains critical. Although perfectly precise definitions may remain elusive, prioritizing these discussions now serves a dual purpose: establishing verification frameworks while simultaneously discouraging the premature development of high-risk autonomous weapons that would inherently resist verification. Compute governance provides a quantifiable and verifiable metric that can facilitate regulatory enforcement mechanisms for AI systems (Sastry et al., 2024). While applying compute-based governance to general-purpose frontier AI models remains contentious due to potential adverse effects on innovation, privacy, economic competitiveness, and power distribution, we propose that a domain-restricted application, specifically to military AI systems, may offer a more viable pathway for governance. This domain-specific approach could address dual-use concerns while avoiding broader innovation constraints. We hypothesize that in military applications, compute scaling exhibits a stronger correlation with specific risk factors than in general AI systems. To validate this hypothesis, we propose empirical research investigating how computational resource allocation affects

capabilities associated with autonomous weapons systems, tactical decision-making, and strategic planning. Inspired by frameworks against tampering (White et al., 2012), we advocate the development of a tamper-resistant safeguards (Tamirisa et al., 2025), and an infrastructure capable of monitoring and verifying compute usage in military AI development across participating nations. This is similar to past efforts in nuclear monitoring and verification (Fournier et al., 2016; Smartt, 2022). If research confirms a robust relationship between computational constraints and risk reduction in military AI applications, such an infrastructure could provide the technical foundation for verifiable arms control agreements. This could potentially establish compute limitations as a technically enforceable mechanism for mitigating specific military AI risks while preserving beneficial AI development in other domains.

## 4.2. Cooperative AI between Adversaries

As military artificial intelligence advances toward complex multi-agent scenarios involving both human and AI agents, new failure modes and systemic risks emerge that require innovative mitigation strategies (Hammond et al., 2025). We propose that research focused on adversarial cooperation mechanisms offers a promising approach to address these challenges. Drawing parallels from Cold War stability theory, adversaries possessing catastrophic capabilities must establish cooperative frameworks to prevent mutually destructive outcomes. This necessitates research into collective decision-making paradigms, including reinforcement learning from collective human feedback and simulated multiparty negotiations (Conitzer et al., 2024). Such approaches enable the formal modeling of cooperative equilibria even under conditions of strategic competition. The concept of epistemic communities provides a valuable framework for implementing these cooperation mechanisms. These communities, which are networks of experts sharing domain knowledge, validation methods, and common objectives (Adler, 1992), can incorporate diverse stakeholders beyond scientific experts, including military strategists, diplomatic corps, and civilian leadership (Cross, 2013). As frontier AI models demonstrate increasingly sophisticated cooperative capabilities (Zhang et al., 2023; Luo et al., 2024), these systems could function as participants within international epistemic communities specifically designed to manage conflicting objectives while maintaining stable arms control regimes. This integration represents an understudied research direction that merits substantial investigation. Future work should build upon existing research in multi-objective optimization for multi-agent systems (McPartland et al., 2005; Hu et al., 2023), mechanisms for sustaining cooperation in competitive environments (Leibo et al., 2017; Piatti et al., 2024), and communication structures within epistemic networks (Haas, 1992; Zollman, 2007). Developing formal

frameworks and empirical validation methods for human-AI epistemic communities should be prioritized to establish robust cooperative mechanisms before advanced military AI systems become widely deployed.

## 4.3. Mitigating Control Loss

Finding a way to mitigate gradual disempowerment will be difficult because it will involve subtle decreases in human control that might not be detected before human authority has been ceded. In Kulveit et al. (2025), the research priorities for mitigating gradual disempowerment are (1) distinguishing between beneficial AI augmentation of human capabilities and problematic displacement of human influence, (2) finding the key thresholds or tipping points in these systems beyond which human influence becomes critically compromised, and (3) measuring the effectiveness of various intervention strategies. Even when restricting ourselves to military AI systems, gradual disempowerment requires more exploratory research that develops a formal framework that clearly defines it. To combat disempowerment in military AI systems, more research should investigate how to strategically make the most of the cognitive abilities that humans can still do better than AI models (Gigerenzer & Goldstein, 1996; Gigerenzer, 2022). As the concept of gradual disempowerment becomes clearer, detection methods similar to time series forecasting or longitudinal studies might infer disempowerment before the worse impacts occur.

## 5. Alternative Views

A core premise of our position is that arms control has been, and will continue to be, a successful form of policy and diplomacy that mitigates catastrophic risk. This assumption may be incorrect. As previously stated, the success of any arms control agreement is contingent upon all participants cooperating to make certain that the goals are satisfied. Like past work in arms control, this would involve AI researchers, policy analysts, and policymakers from different nations to negotiate how autonomous weapons are regulated.

One common criticism of arms control is that it is too compromising to a nation's security. From this perspective, arms control introduces the difficult challenge of balancing transparency and security. The thinking is that geopolitical instability can also arise if a nation reveals military vulnerabilities that could be exploited by an adversary (Coe & Vaynman, 2020). Others also argue that in some cases, successful arms control may actually increase the likelihood of war by impairing the restraining influence of the balance of power (Schofield, 2000). Lastly, some claim that the entire endeavor of arms control is futile in changing conflicting interests (Codevilla, 2015).

The Biological Weapons Convention (BWC) (United Nations Office for Disarmament Affairs, 1972) offers a compelling case study in arms control verification challenges that parallel those facing AI systems. Three critical factors have undermined BWC verification efforts: first, the diverse stakeholder landscape spanning national governments and private industry; second, the global democratization of research capabilities; and third, the predominantly dual-use nature of biological research technologies. These factors have rendered a formal verification regime politically unattainable despite decades of negotiation. Military AI systems likely present even greater verification obstacles given the intangible nature of software, the rapid development cycles, the complex supply chains, and the technical difficulty in distinguishing between civilian and military AI applications. Without credible verification mechanisms, arms control frameworks may provide limited security benefits for military AI systems.

At present, it seems like an insurmountable hurdle for all nations with advanced AI and military capabilities to willingly make military AI systems more transparent. This political reality is highlighted by the recent expiration of the New START treaty. These hurdles, however, make the risks of military AI more urgent, not less. Global security experts acknowledge geopolitical realities but still believe in substantive arms control agreements (Albertson, 2026; Dumbacher, 2026). The public's desire for clear, evidence-based explanations of reforms and policy shows that trust can be regained if public institutions make the consistent effort to improve current political perceptions. We believe that some compromises must be made between all parties to ensure the catastrophic global risk from autonomous weapon systems is minimized. We also presume that if the aforementioned drawbacks of AI arms control are currently too risky, the best course of action is to proceed cautiously with AI advancements and engage in conversations to address current and future issues to identify pathways to prevent destabilizing military AI advancements and events.

## 6. Discussion towards Scientific Policy Consensus

AI researchers must develop technical solutions supporting future arms control objectives (trustworthiness, verification mechanisms, and transparency) to minimize risks posed by military AI systems. We propose a deliberate, measured approach that integrates arms control compliance (Schörnig, 2022). Major military powers are developing AI weapons systems at a rate that creates a dangerous imbalance where technological development outpaces technical safety research (Simmons-Edler et al., 2025). Deploying such systems without robust arms control considerations undermines decades of diplomatic progress in weapons risk mitigation. To effectively address these unique challenges, new AI governance institutions are likely necessary (Dafoe, 2018). The path toward more immediate progress would be the development of venues that foster open dialogue between experts from many disciplines. The Pugwash Conferences on Science and World Affairs is a famous example of scientists doing their part in diminishing nuclear arms in international politics (Nobel Prize Outreach, 1995). Bommasani (2025) advocates for prestigious AI conferences to lead on AI policy consensus and suggests the Intergovernmental Panel on Climate Change as a model for assessing such agreements and negotiations (De Pryck, 2021). We advise AI experts who wish to engage in this conversation to learn how arms control experts view agreements (Shah, 2013), strategic stability (Gerson, 2013), verification (Fortakov, 1998), and deterrence (Mazarr, 2018). AI experts new to these concepts, like many in the general public (Allison et al., 2022), may question the claim by some national security specialists that catastrophic retaliation capabilities are necessary for deterrence (Jenkins et al., 2018). To facilitate constructive collaborations, both groups of experts must clearly describe their goals, agree on a set of facts, be transparent about disagreements, and ensure shared understanding of any joint progress and accomplishments.

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
