# OpenReview forum: "Position: AI Researchers Must Help Lead Arms Control to Mitigate Military AI Risks"
_ICML.cc/2026/Position_Paper_Track — ICML 2026 Position Paper Track regular_

### Official Review · Reviewer_tEqz · 2026-03-11

**Significance:** 3
**Argument Clarity:** 3
**Ethics Flag:** Yes
**Rating:** 5
**Confidence:** 3

**Questions:**

Typos, etc.:
- L248: enhance -> enhanced
- Section 3.3: first sentence does not parse.
- Many links in the bibilography overflow and should be fixed.
- Do you have permission to reproduce Figure 1?

**Alternative Views Section:**

Yes

**Compliance With Llm Reviewing Policy A Conservative:**

Affirmed.

**Discussion Potential:**

3

**Ethical Review Concerns:**

An ethics review may be required for this paper given it approaches the topic of military applications of AI directly. However, given it is a position paper rather than one proposing AI methodology, I do not believe the contents of this paper per se have the potential to be misused. In my view, the position being advocated in this paper is carefully considered, and a responsible one overall; potentially steering the community into thinking about such implications in their own work.

**Ethics Review Area:**

["Inappropriate Potential Applications & Impact (e.g., human rights concerns)"]

**Final Justification:**

I maintain my original recommendation of Accept. The authors engaged meaningfully with my comments and responded with proposed revisions to the work to address the main weaknesses I identified (clarifying the position; accessibility to the AI community). I have decreased my confidence rating to reflect my lack of familiarity with the literature beyond AI (to international relations, AI governance, and war studies), which plays an important role in the assessment of the paper, and on which I cannot provide an informed opinion.

**Paper Summary:**

This paper considers the proliferation of the use of AI for military applications. It discusses the risks associated with this, and proposes that arms control is a suitable approach for mitigating said risks. The paper supports this position primarily by analogies to nuclear technology, which fundamentally shaped today's military doctrines. The authors discuss the challenges in employing arms control for AI, some of which are technical and relate to understanding and ensuring safety in today's AI systems, while other challenges stem from the difficulties of global cooperation between military powers.

**Position:**

Yes

**Position In Title:**

Yes

**Related Work:**

3

**Strengths And Weaknesses:**

## Strengths
S1. The paper is very well-written and well-structured.

S2. The paper manages to draw on both AI and international relations / war studies literatures in a deep and meaningful way, providing insightful analysis and synthesis while remaining quite accessible to an AI audience.

S3. The paper discusses a topic of major importance in today's geopolitical climate.

## Weaknesses
W1. The position behind the paper is slightly unclear: the title and statement suggests that AI researchers should *lead* arms control for AI in military applications. However, it does not actually support this point. Instead, I read it more as supporting the notion of employing arms control methodologies for this purpose overall. The notion that AI researchers can lead this initiative would be challenging to support as governments, military leaders, and the private defense sector have much more power in shaping concrete decision-making, and for initiating this collaboration, than AI researchers.

W2. A few things could be clarified to make this work more accessible to an AI audience:
- What is the "secure second-strike policy" referred to in 2.1?
- At a high level, what do the treaties in L117-120 impose on signatory countries? It would be helpful to understand what type of restrictions countries have voluntarily signed up to in the past, so as to contextualise how challenging such agreements would be for AI.

**Support:**

3

---

> ### Author Rebuttal · Authors · 2026-03-31
>
> Thank you for the generous and thoughtful review. We are glad the paper's synthesis of AI and international relations literatures came through as accessible and insightful, and we appreciate your careful attention to both substance and presentation.
>
> On W1 (AI researchers "leading"): This is a fair and helpful clarification. We agree that governments, military leaders, and the defense sector hold decision-making power. Our argument is that without deep technical input from the AI research community, those decision-makers will lack the tools to craft enforceable agreements. We will soften "lead" to “help lead” to better reflect this collaborative but technically indispensable role. For a more thorough explanation, please read our response to reviewer B7qR.
>
> On W2 (accessibility): Thank you for flagging these. We will add brief, accessible explanations of "secure second-strike capability" (the ability to absorb an initial nuclear attack and still retain enough surviving forces to retaliate, which underpins deterrence stability). Regarding treaties like New START, at a high level, both nations must set numerical limits on their nuclear warheads and delivery systems. Defining a quantity (or quantities) that represents catastrophic AI risk will be another question the AI community must ask itself.
>
> We confirm we have permission to use Figure 1 and document the permission in the final version.
>
> Proposed changes: We will fix all noted issues (L248, Section 3.3 first sentence, bibliography overflow). We will also change the title from “AI Researchers Must Lead Arms Control to Mitigate Military AI Risks” to ”AI Researchers Must Help Lead Arms Control to Mitigate Military AI Risks”. Also explain “secure second-strike capability”.
>
> Thank you again for the constructive and encouraging review.

---

> > ### Author Rebuttal · Reviewer_tEqz · 2026-04-01
> >
> > Many thanks for engaging with my comments, I agree with the proposed changes and believe they will strengthen the paper.

---

### Official Review · Reviewer_B7qR · 2026-03-12

**Significance:** 3
**Argument Clarity:** 3
**Rating:** 4
**Confidence:** 3

**Questions:**

See Strengths and Weaknesses.

**Alternative Views Section:**

Yes

**Compliance With Llm Reviewing Policy A Conservative:**

Affirmed.

**Discussion Potential:**

3

**Final Justification:**

The authors' response has addressed my concerns, especially regarding the analogy of nuclear weapons. So, I will keep my positive rating for the paper.

**Paper Summary:**

This paper argues that the rapid integration of AI into military systems presents novel and catastrophic risks that current arms control frameworks are unprepared to handle. Drawing parallels from the history of nuclear arms control, the authors identify specific dangers such as AI-driven conflict escalation, deceptive "alignment faking", and the gradual disempowerment of human decision-makers. They contend that technical solutions for verification and transparency are critically lacking. Therefore, the paper's central thought is that AI researchers must take a leading role in collaborating with arms control experts to develop the technical and diplomatic tools necessary to mitigate these emerging threats and prevent a destabilizing AI arms race.

**Position:**

Yes

**Position In Title:**

Yes

**Related Work:**

3

**Strengths And Weaknesses:**

**Strengths**

- This paper integrates historical nuclear policy with AI safety research by using the mature framework of nuclear arms control as an example, which provides a concrete and compelling context for AI researchers who may be unfamiliar with these diplomatic concepts, grounding abstract future risks in real-world precedent.

- The authors pinpoint specific, empirically demonstrated risks of current AI models in military contexts by illustrating that LLMs exhibit escalatory behavior in wargames.

**Weaknesses**

- This paper relies on the nuclear arms control analogy to form the position, which may obscure unique challenges posed by AI. Unlike nuclear weapons, AI is digital, dual-use, developed primarily by private companies, and evolves at an extremely fast pace. The authors briefly acknowledged these differences but did not deeply explore them. These differences may limit the applicability of historical solutions.

- The paper presents alternative views, such as the difficulty of achieving transparency between adversaries, but it does not deeply engage with the political obstacles to its proposed cooperation. The assumption that rival nations would agree to share model weights, submit to international compute monitoring, or cooperate on "adversarial cooperation mechanisms" seems too optimistic.

====

After rebuttal: The authors' response has addressed my concerns, especially regarding the analogy of nuclear weapons.

**Support:**

3

---

> ### Author Rebuttal · Authors · 2026-03-31
>
> We thank the reviewer for their constructive response. We address the two main concerns directly.
>
> On the analogy obscuring unique AI challenges:
> We want to reframe this concern. The analogy is not meant to claim that AI weapons are identical to nuclear weapons. It is meant to provide an intuitive entry point into a problem space that currently lacks one. There is no consensus on what catastrophic AI looks like. Nuclear weapons provide a shared reference point that makes the stakes immediately visceral in a way that discussions of dual-use risk or misalignment simply do not, at least not yet.
>
> That said, the differences the reviewer raises are real, and we want to address them specifically:
>
> 1. AI evolves at an extremely fast pace: This is true, but outpacing governance is not a challenge unique to AI. It is a recurring feature of arms races. Nuclear arsenals went from roughly 300 weapons globally in the early 1950s to over 30,000 by the mid-1960s. The speed was different in absolute terms, but the governance challenge was structurally identical: technology outran diplomatic frameworks, and the world came perilously close to catastrophe (e.g. Cuban Missile Crisis). The lesson from nuclear history is not that governance kept pace, but that governance eventually caught up and that earlier action would have been far better. This is precisely our argument for AI.
>
> 2. AI is developed primarily by private companies: This is a difference in degree, not in kind. The nuclear weapons complex involved extensive private-sector participation (like contractor-operated national laboratories). Arms control verification activities derived from nuclear weapons treaties have been integrated into the CWC and BWC, both of which has public and private sector participation, in line with what is seen with AI. More importantly, private companies developing AI remain subject to government pressure through export controls, compute regulations, and national security directives. We have seen this repeatedly in recent years, with governments imposing chip export restrictions and pressuring frontier AI labs on safety practices. The private-sector development of AI does not place it beyond the reach of arms control; it simply changes the mechanisms through which control is exercised.
>
> 3. AI is digital and dual-use: While nuclear energy is dual-use, we acknowledge this complicates verification. But the paper's call for technical research into verification (Sec. 5.2) is precisely motivated by this challenge. We do not claim existing nuclear verification protocols can be directly transferred; we argue that the structure of how the nuclear arms control community developed verification (technical experts working alongside diplomats to create novel inspection regimes) should be replicated for AI.
>
> On political obstacles and optimism about cooperation:
> We do not assume rival nations would agree to share model weights or submit to international compute monitoring. Section 4 of our paper acknowledges these difficulties, and the research directions in Section 5 are framed as problems to be solved. The purpose of the paper is to argue that AI researchers should engage with these problems, not to claim they are already solved. We will make it clear that these are just examples of verifying model capabilities, and clearly state that these are not politically realistic now. However, the difficulty of the political obstacles strengthens our position that technical experts need to be involved early, because the feasibility of any future agreement will depend on whether technically sound verification mechanisms exist. We appreciate the reviewer pointing out that cooperation can fail. Even absent formal agreements, space-based ISR using high-resolution electro-optical imagery and thermal infrared sensing offers a technically viable path for data center monitoring, directly paralleling how technical underpinned nuclear arms control verification long before cooperative inspection regimes existed. The infrastructure required for large-scale military AI training has a substantial and detectable physical footprint. Hence, even in the least optimistic diplomatic scenario, compute verification remains feasible, but more work needs to be done by the AI community for more effective policy.
>
> Proposed revision: We will strengthen the alternative views in Section 4 to highlight the political realities and more explicitly engage with the unique properties of AI that complicate arms control (speed of development, dual-use nature, etc.), while arguing that these differences make the case for early technical engagement more urgent, not less. We will also add the non-nuclear dangers for each risk in section 3.
>
> We respectfully note that political obstacles, while real and important to acknowledge, are by nature dynamic and evolving. We also commit to adding non-nuclear dangers to strengthen our narrative. We hope the proposed revision will address reviewer’s concerns.

---

> > ### Author Rebuttal · Reviewer_B7qR · 2026-04-01
> >
> > Thanks for the authors' rebuttal. My concerns have been addressed, and I am happy to increase my score.

---

### Official Review · Reviewer_ZVtx · 2026-03-13

**Significance:** 4
**Argument Clarity:** 3
**Rating:** 5
**Confidence:** 5

**Questions:**

-In section 2.1, what (if anything) has changed from 2022 to 2026 in US DoD nuclear and deterrence posture? It has been an eventful four years in geopolitics since then, including the expiration of NEW-START this year, the last remaining nuclear arms limitation treaty. While to my knowledge there is a stated desire by both parties to replace it with a new bilateral or trilateral (including China) nuclear arms treaty, it seems relevant to address recent lapses in arms restriction and confirm that the core logic remains long-term valid.

-In section 2.3, what would examples of AI CBM's be? The discourse around transparency in frontier AI models is substantial, is there a CBM-type argument in favor of transparency for geopolitical stability, for example?

-In 5.1, what risk factors is "We hypothesize that in military applications, compute scaling exhibits a stronger correlation with specific risk factors than in general AI systems" referring to? This seems like a point worthy of some elaboration.

-Also in 5.1, how do the technical distinctions between (for example) on-platform visual target identification and selection AI for low cost UAVs and command-advising LLMs that run in datacenters in rear areas impact questions of compute constraints? The former class of system inherently needs less compute to train and operate, but can present similar (or greater, if deployed recklessly and in diplomatically tense situations where misbehavior could cause conflict escalation) risks. Is some form of breakdown by system category with different thresholds defined for each needed?

-In 5.3, discussion/consideration of the concepts of meaningful human control (MHC) versus appropriate human judgement (AHJ), which are the dominant frameworks for addressing similar issues of maintaining human authority over military AI systems in deployment, would benefit this section. Some of the questions that those frameworks provoke (how much control is needed to be "meaningful?" what compromises in system capability must be made to maintain that? under AHJ, where does human responsibility for the actions of an AI system lie? etc) are well aligned with the questions posed in this section.

-In 5.4, in addition to purely technical discussion on military AI risks, what about interdisciplinary conferences including both AI technical experts as well as legal, policy, and military experts on AI for the military domain? As an example, the annual REAIM conference aims to foster such dialogue as part of establishing international consensus on regulating military AI, but features a relatively small technical expert presence. What can be done to encourage more engagement between technical and nontechnical experts on this topic?

-As a general suggestion, Simmons-Edler et al, 2025 (https://arxiv.org/abs/2505.18371) covers some closely related ground on the need for AI technical expertise in developing military AI regulation, and while not redundant it might be useful context to discuss or compare to.

**Alternative Views Section:**

Yes

**Compliance With Llm Reviewing Policy A Conservative:**

Affirmed.

**Discussion Potential:**

3

**Final Justification:**

The rebuttal proposes several reasonable changes which will largely address the issues with clarity and focus I raised, which removes the reservations I had on wholeheartedly recommending the paper for acceptance.

**Paper Summary:**

This paper argues the need for AI researchers to take the initiative in developing and promoting arms control frameworks and mechanisms to address the risks of military AI, particularly in the nuclear domain. It describes a number of potential risks posed by military AI, and argues by analogy to nuclear arms control that many of these risks can be mitigated by international agreements on arms limitations. The paper then concludes with some discussion of directions for further work to enable such arms control agreements in the future.

**Position:**

Yes

**Position In Title:**

Yes

**Related Work:**

3

**Strengths And Weaknesses:**

Overall, this paper makes a clear and salient argument on a topic that is highly relevant to the ICML community (and has only grown moreso since the submission deadline, with new reports of frontier AI model usage in Venezuela and Iran). The writing is largely clear, and the arguments are overall reasonable.

The main limitation/weakness of this work is the narrow focus it places on AI in the nuclear domain. While the position in favor of technically-driven regulation is not limited to nuclear AI, it is the only specific military application domain discussed, and while this does help strengthen the analogy to historical nuclear arms control agreements, it does mean there is not a lot of support or focus given to non-nuclear military AI arms control and associated issues. Similarly, it would strengthen the paper to have a clear outline of what the technical barriers to arms control are for specific categories of military AI (for example, nuclear command and control models versus tactical strategic command-advising LLMs versus on-platform kinetic targeting AI models for battlefield use).

In addition, I found the paper's structure slightly confusing. In particular, section 5 is presented as directions for future work, but contains much of the discussion of the technical barriers to arms control in the paper. I think it would make the paper more persuasive if these directions and challenges were discussed in section 3 alongside their corresponding risks, or immediately following them.

Despite these issues, the salience of the issue to the community in the present moment and the generally well constructed arguments lead me to recommend acceptance, though the paper would be strengthened by addressing these weaknesses (and some of the points raised in the questions section below).

**Support:**

3

---

> ### Author Rebuttal · Authors · 2026-03-31
>
> We thank Reviewer ZVtx for the thoughtful and detailed review and for the highly constructive suggestions. We address each point:
> On narrow focus on the nuclear domain:
>
> We appreciate this concern, and we refer to our discussion in the response to Reviewer 8uuj above regarding the deliberate nature of this choice. We agree that the proposed mapping table connecting nuclear-scenario risks to non-nuclear military AI categories would strengthen the paper without diluting its core argument.
>
> Proposed Changes:
> We agree that integrating the technical barriers from Section 5 into Section 3 alongside their corresponding risks would improve readability and persuasive flow. We will switch sections 4 and 5 to improve narrative flow. For each AI risk in section 3 (like alignment faking), we will add non-nuclear example of potentially dangerous use. We will add the APLN analysis demonstrates our scenario's grounding in current geopolitical reality. We will also sharpen the research directions which draw on nuclear deterrence’s past successes, generalize across non-nuclear weapons control, update section 2 to mention New START’s expiration, and mention the need for discussions on meaningful human control.
>
> On New START expiration:
> This is an excellent and timely point. The expiration of New START in 2026 is directly relevant. As Albertson (2025) argues (https://www.atlanticcouncil.org/in-depth-research-reports/issue-brief/new-start-might-be-dead-but-legally-binding-arms-control-isnt/), the demise of New START does not invalidate the logic of legally binding arms control; rather, it shifts the burden toward new frameworks that can accommodate emerging technology categories including AI. If anything, this development strengthens our core argument: the erosion of existing arms control infrastructure makes it more, not less, urgent for technical communities to develop the verification and transparency tools that future agreements will require. We will incorporate this discussion into Section 2.1.
> On AI CBMs: We will make it clear that AI confidence-building measures for military AI is needed, which would be helpful in building pre-notification of large-scale military AI exercises, shared incident reporting frameworks for autonomous system malfunctions, and transparency measures around frontier model deployment in military contexts (drawing the parallel to CBMs in the nuclear context such as launch notification agreements).
>
> On compute thresholds across system categories: This is a valuable point. We will add a brief discussion acknowledging that risk-relevant compute thresholds differ substantially between system categories (on-platform edge AI vs. datacenter-scale advisory systems) and that a workable framework may require category-specific thresholds rather than a single universal metric.
>
> On MHC/AHJ frameworks: We will mention the need for meaningful human control and appropriate human judgment to Section 5.3, as these frameworks provide the doctrinal and legal grounding for our more abstract discussion of disempowerment.
>
> On interdisciplinary engagement: We fully agree and will add discussion of venues like REAIM and the need for greater technical expert participation. We will also reference Simmons-Edler et al. (2025) as suggested.
>
> We are grateful for these suggestions and believe they will substantially strengthen the paper.

---

> > ### Author Rebuttal · Reviewer_ZVtx · 2026-04-03
> >
> > Thanks for the positive response! I think the proposed changes will strengthen the paper significantly, and will increase my score accordingly.

---

### Official Review · Reviewer_8uuj · 2026-03-19

**Significance:** 2
**Argument Clarity:** 3
**Rating:** 4
**Confidence:** 4

**Questions:**

1. In the first paragraph of Sec. 2.3, it is stated that [Simmons-Edler et al., 2024] "lists many airborne, ground, and naval autonomous weapons systems, as well as autonomous weapons systems of command." Can you elaborate further on what makes these **weapons** systems? Specifically, in any of these systems, is the autonomy pipeline able to use lethal force without human authorization?
2. In Sec. 5.1, can you clarify the statement "[w]e hypothesize that in military applications, compute scaling exhibits a stronger correlation with specific risk factors than in general AI systems"? In addition, can you elaborate on what the proposed "empirical research investigating how computational resource allocation affects capabilities" would look like in practice?
3. Can you state the central hypothesis of Sec. 5.2? Specifically, in the proposed "adversarial cooperation mechanisms," are adversarial groups of humans doing the cooperating, or are adversarial AI agents cooperatively interacting in some way?
4. In Sec. 5.3, is the proposed direction to elaborate the work [Kulveit et al., 2025] for the specific military AI context? Or is the proposed direction to figure out what disempowerment means in the military AI context?

**Alternative Views Section:**

Yes

**Compliance With Llm Reviewing Policy A Conservative:**

Affirmed.

**Discussion Potential:**

2

**Final Justification:**

The paper argues effectively for its position. The author rebuttal addressed several of the concerns outlined in my review.

**Paper Summary:**

The position of the paper is that AI researchers should lead military AI arms control research in conjunction with arms control experts. An overview of historical arms control in the context of nuclear arms is provided. The risks of future military AI use is illustrated via a running thought experiment of using LLMs in nuclear conflicts. Within this context, four main risks are discussed in light of recent LLM research: conflict escalation, alignment faking, human disempowerment, and handling catastrophic failures of LLM decision-making. Potential research directions for addressing these risks are outlined.

**Position:**

Yes

**Position In Title:**

Yes

**Related Work:**

3

**Strengths And Weaknesses:**

**Strengths**

The paper provides good historical background on nuclear arms control, which motivates the running AI nuclear arms control example well. The nuclear arms control example is rhetorically effective, because the stakes are high: malfunctioning AI systems increase the risk of nuclear war across all the examples. The specific scenarios described based on recent LLM research (escalation, alignment faking, disobedience) are plausible within the context of the running example, and the catastrophic nature of the associated risks are clear in the nuclear setting. Finally, the general importance of continuing research into military AI arms control is clear.

**Weaknesses**

Though the nuclear arms control thought experiment is rhetorically effective, it is unrealistic. Without substantiating evidence to the contrary, it seems unlikely that AI will be used for this highest-of-all-stakes decision-making at any point in the foreseeable future. While the authors explicitly state that the nuclear arms control example is used for clarity of argumentation and to draw parallels with historical nuclear arms control, it also results in limited discussion of concrete examples of how AI is being used or being proposed or speculated to be used in real military applications. Focusing on the nuclear arms control example instead detaches the argument for the position from immediate, realistic concerns associated with use of AI in military contexts, weakening the overall argument. Finally, though the general research directions outlined are important, the concrete directions mentioned are vague (see Questions), weakening the potential of the paper to stimulate follow-on work.

**Support:**

2

---

> ### Author Rebuttal · Authors · 2026-03-31
>
> We appreciate the reviewer's constructive feedback. We want to directly engage with the central concern that the nuclear framing is unrealistic and that it detaches the argument from immediate, realistic concerns.
>
> The nuclear focus is a deliberate rhetorical strategy, not an oversight. Our approach mirrors that of the AI 2027 report, which painted a vivid, specific scenario of AI development rather than attempting to catalog every possible trajectory. That report was effective precisely because it chose depth over breadth, allowing readers to internalize concrete dangers rather than skim a survey of abstract possibilities. We adopted the same logic. A position paper that attempts to enumerate every category of military AI risk within the page limit would inevitably dilute each argument. The nuclear analogy cuts through genuine ambiguity in ways that general "dangerous AI capabilities" framing cannot. There remains significant, active disagreement within the AI safety community about what catastrophic AI even looks like. Does it mean misaligned superintelligence? Autonomous drone swarms? The lack of consensus is itself evidence that abstract framings of AI danger struggle to galvanize coordinated action. Nuclear weapons, by contrast, tap into a deep, intuitive, and near-universal human fear. Everyone understands what nuclear exchange means. By grounding AI risks in this familiar context, we bypass the ambiguity that has stalled progress on AI governance and make the dangers immediately legible to policymakers, military strategists, and AI researchers alike.
> Critically, the nuclear scenario is not as unrealistic as the reviewer suggests. Recent analysis from the Asia-Pacific Leadership Network (https://www.apln.network/analysis/commentaries/ai-is-quietly-reshaping-nuclear-risk-in-south-asia) documents how AI is already quietly reshaping nuclear risk in South Asia, with nations integrating AI into surveillance, early warning, and strategic decision-support systems that indirectly but materially affect nuclear posture. The pathway from "AI advises on nuclear-adjacent decisions" to "AI is embedded in nuclear command and control" can be accomplished through escalation pressures beyond human control that encourage increasing AI military capabilities. The thought experiment in our paper is designed to illuminate the endpoint of trends that are observable today. Hence, we must guard against current military AI dangers and medium-term nuclear AI dangers at the same time.
> Regarding the specific questions:
> 1. On weapons systems from Simmons-Edler et al., 2024: They say the UN passed a resolution calling for strict regulation on AWS, and for humans to stay in control of lethal decision-making. The authors of that paper are likely better able to answer this question.
> 2. On compute scaling hypothesis (Sec. 5.1): Our hypothesis is that military applications create tighter coupling between compute and specific dangerous capabilities (e.g., targeting accuracy, strategic planning depth, speed of autonomous response) compared to general-purpose AI, where compute scaling improves broad benchmarks. In practice, empirical research could involve measuring how model performance on military-relevant tasks (wargame escalation rates, targeting precision, adversarial robustness) scales with compute, and comparing these scaling curves to general capability benchmarks. Because this is a difficult problem, we ask the entire research community to help solve this problem.
> 3. On adversarial cooperation mechanisms (Sec. 5.2): We assume human-AI teams where each team tries not to escalate the situation. The difficulty is when the AI components of opposing teams start to escalate in the absence of humans during certain processes.
> 4. On Sec. 5.3: The proposed direction is more of the latter: first, to define what disempowerment means in the military AI context specifically (where chains of command and rules of engagement create a structured decision hierarchy unlike civilian settings), and then to extend the framework of Kulveit et al. (2025) to address this military-specific definition. That way, it will be easier to clearly look for specific instances of disempowerment.
> Proposed revision: For each AI risk in section 3 (like alignment faking), we will add non-nuclear examples of potentially dangerous use. We will add the APLN analysis demonstrates our scenario's grounding in current AI escalation realities. We will also sharpen the research directions which draw on nuclear deterrence’s past verification successes and generalize across current AI weapons control, and improve section 5 by framing the section as a call for the AI community to contribute their own ideas.
> We respectfully ask the reviewer to consider whether the narrowness they identify is in fact the paper's core methodological choice rather than a weakness, and whether the new reference is a real-world military application.

---

> > ### Author Rebuttal · Reviewer_8uuj · 2026-04-03
> >
> > Thanks to the authors for their response, which has helped address several of my concerns. I have accordingly increased my score. Some additional suggestions/questions for the authors:
> > 1. Regarding response 1 from the rebuttal: if you do not know what you are referring to when you write "many airborne, ground, and naval autonomous weapons systems, as well as autonomous weapons systems of command," it would be wise to either find out or to remove the reference.
> > 2. The "[r]ecent analysis from the Asia-Pacific Leadership Network" that you cite in your rebuttal sounds like it could mitigate the implausibility of your nuclear arms control example -- consider including a discussion in the revision.
> > 3. In your response you mention "escalation pressures beyond human control that encourage increasing AI military capabilities." There is a cognitive leap lurking here: what are the "pressures beyond human control" that could lead to AI gaining lethal force capabilities? This leap is also made in the submission in the nuclear arms control context. Providing a plausible rationale for how this might occur would strengthen the paper.

---

### Decision · Program_Chairs · 2026-04-30

**Decision:**

Accept (regular)

**Comment:**

The paper’s main strengths are that it tackles an urgent topic, connects AI safety with arms-control and international-relations thinking in a way that is accessible to the ICML audience, and offers a concrete research agenda rather than only a warning. The main remaining limitations concern scope and realism: the nuclear framing is rhetorically strong but narrow, and the final version should more clearly ground its argument in present-day military AI deployments and sharpen the technical research directions. The overall review is positive, and all four reviewers indicated that the rebuttal resolved their main concerns. I believe the paper merits acceptance as a position paper.